# A Virtual Power Plant Solution for Aggregating Photovoltaic Systems and Other Distributed Energy Resources for Northern European Primary Frequency Reserves

Rakshith Subramanya [1,*], Matti Yli-Ojanperä [1], Seppo Sierla [1], Taneli Hölttä [1], Jori Valtakari [2] and Valeriy Vyatkin [1,3,4]

1    Department of Electrical Engineering and Automation, School of Electrical Engineering, Aalto University, FI-00076 Espoo, Finland; matti.yli-ojanpera@aalto.fi (M.Y.-O.); seppo.sierla@aalto.fi (S.S.); taneli.holtta@aalto.fi (T.H.); valeriy.vyatkin@aalto.fi (V.V.)
2    Siemens Osakeyhtiö, 02600 Espoo, Finland; jori.valtakari@siemens.com
3    SRT, Luleå University of Technology, 97187 Luleå, Sweden
4    International Research Laboratory of Computer Technologies, ITMO University, 197101 St. Petersburgh, Russia
*    Correspondence: rakshith.subramanya@aalto.fi

**Abstract:** Primary frequency reserves in Northern Europe have traditionally been provided with hydro plants and fossil fuel-burning spinning reserves. Recently, smart distributed energy resources have been equipped with functionality needed to participate on frequency reserves. Key categories of such resources include photovoltaic systems, batteries, and smart loads. Most of these resources are small and cannot provide the minimum controllable power required to participate on frequency reserves. Thus, virtual power plants have been used to aggregate the resources and trade them on the frequency reserves markets. The information technology aspects of virtual power plants are proprietary and many of the details have not been made public. The first contribution of this article is to propose a generic data model and application programming interface for a virtual power plant with the above-mentioned capabilities. The second contribution is to use the application programming interface to cope with the unpredictability of the frequency reserve capacity that the photovoltaic systems and other distributed energy resources are able to provide to the frequency reserves markets in the upcoming bidding period. The contributions are demonstrated with an operational virtual power plant installation at a Northern European shopping center, aggregating photovoltaic Primary Frequency Reserves resources.

**Keywords:** solar power; virtual power plant; application programming interface; primary frequency reserve; frequency containment reserve; demand response; forecasting; machine learning; neural network





## 1. Introduction

One significant trend in the transition to renewable energy is that the need for PFR (Primary Frequency Reserves) increases due to the penetration of Photovoltaic (PV) and wind power generation, while conventional providers of PFR, namely fossil fuel burning plants with spinning reserves, are being replaced by renewable energy sources [1]. In response to this trend, there is a booming body of research on innovative solutions for providing PFR with smart loads (e.g., [2,3]), wind power generators (e.g., [4,5]), and electric vehicles or other battery storages (e.g., [6,7]).

One major constraint that is often neglected in academic research is that the minimum capacity requirement for the PFR market must be met. For example, in the Finnish PFR markets Frequency Containment Reserve for Normal Operation (FCR-N) and Frequency Containment Reserve for Disturbances (FCR-D), the minimum capacity of one bid is 0.1 Megawatt (MW) and 1 MW, respectively [8]. Thus, aggregation of the PFR providing reserve resources by a Virtual Power Plant (VPP) or similar solution is necessary for all

energy resources that do not by themselves satisfy such requirements. However, real-world VPPs remain proprietary solutions with little publicly available documentation. In order to catalyze research in this area, the goal of this paper is to propose an architecture and data model for a generic VPP application programming interface to encapsulate proprietary details of commercial VPPs.

One major use case for the virtual power plant application programming interface is energy forecasting. Several machine learning forecasting applications relevant to virtual power plants have been proposed for wind power generation [9]; photovoltaic power generation [10]; electric vehicle charging load [11]; electric vehicle battery state-of-charge [12]; battery storage state-of-charge [13]; and Heating, Ventilation and Air Conditioning (HVAC) load [14]. This article demonstrates the use of the virtual power plant application programming interface for one energy forecasting application. However, the contribution of the article is not the forecasting solution. The contribution is the virtual power plant information architecture, data model, and application programming interface, which support the development of energy forecasting applications beyond the one presented in this paper, such as the ones referenced above, as well as other analytics applications exploiting virtual power plant data.

This paper is structured as follows. Section 2 reviews related works and places the currently limited research on photovoltaic PFR into the context of emerging types of PFR. Section 3 presents our proposal for an architecture and data model for a generic VPP application programming interface. Section 4 presents use cases for the VPP application programming interface. Section 5 describes technology choices for implementing the architecture and data model in Section 3 for our case study. Section 6 applies the VPP application programming interface to realize the use cases in the context of an operational virtual power plant installation at a Finnish shopping center, aggregating photovoltaic and other PFR resources. Section 7 discusses the exploitation scenarios of the results. Section 8 concludes the paper.

## 2. Related Research

### 2.1. Primary Frequency Reserves Markets

Technical requirements for PFR originate from an era in which fossil fuel-based solutions were used to provide the PFR. As the power system inertia is impacted by the penetration of wind and photovoltaic generation, authors are calling for shorter response time requirements in the provision of frequency reserves [15,16]. The emergence of new solutions for PFR, such as controllable loads, offers a solution for meeting such requirements [17]. Since the loads can have much faster response time than generators, due to a lack of inertia, proposals have been made to redesign the established market and control mechanisms for PFR. Li et al. [18] redesigned the PFR market mechanism to account for the response time in the valuation of offers. Liu & Du. [19] approached the problem by proposing separate markets for generators and loads participating in PFR, so that the procurement of resources on these two markets is coordinated. However, this article focusses on existing markets, since data from the operation of existing PFR solutions on existing markets is required to solve the research problem.

### 2.2. Novel Solutions for Primary Frequency Reserves

#### 2.2.1. Smart Loads

The development of a smart load capable of providing PFR requires innovation that takes into account the specific characteristics of the load, so that the PFR can be provided without compromising the primary function of the load. Liu et al. [20] proposed an approach for controlling Light-Emitting Diode (LED) luminaires to obtain the PFR while maintaining user satisfaction within acceptable limits. Several different solutions have been presented to aggregate refrigerators while maintaining the temperature of each refrigerator in the desired range [2,21,22]. Zhao et al. [23] proposed a more general solution for thermostatically controlled loads. Wu et al. [3] generalized further to HVAC loads.

Biegel et al. [24] proposed an even more general approach for an aggregator managing on/off loads, but this does not take into account the specific constraints of each type of load. Perroy et al. [25] analyzed diverse industrial loads, capture their constraints for providing PFR, and develop a pooling system for aggregating such loads for the provision of PFR. In contrast, De Carne et al. [26] did not try to tailor a control strategy for any particular type of load but perform load sensitivity identification to determine a PFR control strategy with limited impact to the primary usage of the load. Weckx et al. [27] assumed the availability of utility functions, which capture the inconvenience to end users from using their devices for PFR.

### 2.2.2. Energy Storages

In addition to smart loads, energy storage systems have been proposed as a solution for PFR in a scenario where conventional power generation is replaced by wind and photovoltaic generation [28]. Knap et al. [29] determined the minimum size for the storage to cope with a certain volume of wind power generation. If batteries are not dedicated for the purpose of PFR, probabilistic methods are needed to minimize the risk of PFR interfering with the primary purpose of the battery [7]. Electric vehicles are notable examples of such batteries, and specific solutions have been developed for their use to provide PFR. Izadkhast et al. [6] proposed using the rapid load-shedding capability of electric vehicle chargers in order to rapidly react to frequency deviations, so that as soon as the conventional PFR reserves are activated they will relieve the chargers. Kariminejad et al. [30] proposed a hierarchical control of conventional fossil fuel reserves and electric vehicles to minimize the utilization of electric vehicles for PFR. Srinivasan et al. [31] noted that battery storages are generally too small for PFR unless they are aggregated.

### 2.2.3. Photovoltaic and Other Sources of Renewable Power Generation

The adaptation of wind power generators for the provisioning of PFR is feasible and necessary [32–34]. Several variations of droop control have been proposed for wind generators [4,5,35,36]. Several authors argue that such approaches need to be complemented with battery storage system in the case of wind [37,38] and photovoltaic power generation [39]. Molina-García et al. [40] complement wind generator PFR with PFR capable smart loads. You et al. [41] analyzed the opportunity cost of lost power generation resulting by providing PFR with wind or photovoltaic power generators, and call for economic comparisons with other approaches, such as the use of smart loads and energy storages for PFR.

While wind power applications dominate the research on PFR provisioning by renewable power generators, a few works exist related to hydro power. Saarinen et al. [42] noted that hydro plants are already being used for PFR and analyze the economic tradeoff of providing PFR versus selling power to the spot market. Spitalny et al. [43] proposed small hydro plants developed specifically for PFR. Ahmed et al. [44] noted the possibility of providing PFR from pumped storage hydro.

### 2.3. Managing Primary Frequency Reserves with a Virtual Power Plant

It is important to distinguish between Demand Response (DR) and PFR. In a recent survey on DR, Vardakas et al. [45] identified frequency-controlled DR such as PFR as one exotic type of DR. Thus, it is unsurprising that the great majority of VPP solutions for managing DR focus on other types of DR, namely various price-based or incentive-based programs offered to consumers (e.g., [46–48]). Few researchers address the management of PFR capable resources with a VPP. Abbasi. [49] defines PFR-capable loads as DR loads, to be managed by a VPP doing business directly with a transmission system operator. Srinivasan et al. [31] recognize the need for a virtual power plant managing batteries capable of providing PFR, since the individual batteries are generally too small to participate on such markets directly. Molina-Garcia et al. [50] proposed a specific solution for aggregating several small consumer-owned DR-capable loads to meet PFR market requirements. Although the research on managing PFR with a VPP is currently very limited, if the

research reviewed in Section 2.2 results in practical implementations, there will be a need for VPP-like solutions to aggregate and trade these solutions on PFR markets. In order to trade profitably, forecasts will be required, as discussed in the next section.

The primary users of the virtual power plant application programming interface would be developers of advanced functionalities to the virtual power plant, especially related to participation on PFR markets. For example, PFR market price and asset capacity forecasts could be developed by third parties and integrated to the virtual power plant through the proposed virtual power plant application programming interface. Regarding the applicability of the results to other system integrations, the following areas of further research are identified. Calvillo et al. [51] presented an overview of smart city energy management systems such as power grid simulators; the integration of a virtual power plant through our proposed interface could work as a one-way solution (datasets are extracted as input to the simulator), but not as a two way solution as generally commercial virtual power plant packages cannot be expected to have simulation capability. However, our solution could integrate to provide Key Performance Indicators (KPIs) to an urban control center [52]; the asset operational reliability quantified in use case 1 is an example of a possible KPI. Currently, commercial virtual power plants in general, and in our case study in particular, are focusing on electric systems, but in case of extensions to district heating and combined heat and power production, it would be necessary to assess the applicability of our proposed data models to multicommodity smart energy systems [53]. Under current PFR market participation rules, especially in the case study country Finland, virtual power plants are not expected to coordinate; however, the auction-based PFR market may not be optimal from an energy management perspective, so further research on novel PFR markets could be undertaken based on concepts of interactively cooperating virtual power plants [54]. Finally, one essential system integration for any virtual power plant is the integration to the system managing the distributed energy resources. Examples of such systems to be integrated include microgrid controllers, battery management systems, and building automation systems [55,56]. The Representational state transfer (REST) Application Programming Interface (API) presented in this article is one possible integration technique; however, the industry standard IEC 104 may be better supported by such systems.

### 2.4. Forecasting Problems in the Context of Primary Frequency Reserves

Prior to defining the forecasting problem in further detail, some terms need to be defined. Before providing the definitions, an example is provided. In several European countries, and especially in Northern Europe, PFR are called Frequency Containment Reserves (FCR) [22,25,42,57]. Taking the Finnish FCR markets as a concrete example of national PFR markets, the Transmission System Operator, which operates the market, runs an auction on the previous day in which participants bid certain capacities at certain prices for the hours of the next day [8]. If bids are accepted, the bidder is required to monitor the local grid frequency and activate reserve in the case of frequency deviations according to the technical specification of FCR [58]. It is notable that the Transmission System Operator compensates bidders according to the bids and for their reserve capacity at the market clearing price, regardless of whether any activations occurred [8]. If the reserve could not be provided during frequency deviations, the market rules also specify a penalty payment [8].

The following definitions are used in this article. These are in line with Nordic PFR market operator terminology in general and Finnish PFR markets specifications in particular:

1.  The capacity of PFR reserves is defined as the power that is standing by, ready to be controlled in the event of a frequency deviation. Thus, the capacity is a fixed value for the duration of the bidding period, which is 1 h in the case of Finnish FCR.

    a.  The operator of the PFR market, usually a Transmission system operator, needs to forecast the needed PFR capacity in order to decide how much to procure for the upcoming bidding period.

b.  The participant to the PFR market, for example a Virtual Power Plant operator, needs to forecast the available capacity from the resources under its control, in order to bid profitably, to exploit the PFR capacity as fully as possible, and to ensure that it can honor its bids, should they be accepted by the market operator.

2.  The activation of PFR reserves is the actual change in power consumption or generation that is required in response to a frequency deviation that may occur during the delivery period.

3.  The price of PFR reserves is the price that the market operator pays for accepted bids, regardless of whether activations occurred.

4.  The penalty is the payment that a bidder must pay to the market operator, in case the bid was accepted, and activations occurred, but the PFR reserve could not be provided.

In order to use the PFR reserves profitably on the PFR market, it is desirable to forecast the capacity, activation, and price as well as the likelihood of incurring penalties. Based on such forecasts, intelligent decision making would be possible. However, most publications that discuss forecasting in the context of PFR do not address the items 1–4. One reason for the existence of PFR is to cope with errors in load forecasting, which result in the wrong quantity of electricity generation being procured, resulting in imbalance of electricity production and consumption, which can be managed at delivery time by PFR (e.g., [59–61]). However, much research has already been done on load forecasting (e.g., [62,63]), which is a separate problem from forecasting capacity, activation, price, and penalty as defined above.

A few publications address forecasting the items 1–4. Relevant to 1a are forecasts of PFR capacity required to cope with the variability of photovoltaic power generation [64]. In contrast, Wang et al. [32] forecasted the reserves that can be provided by PFR capable wind power generators, relevant to 1b. Srinivasan et al. [31] considered a virtual power plant managing diverse resources capable of providing PFR. Their approach involves creating a control signal forecast for a model predictive controller, which can be categorized as activation forecasting as defined in 2. Giovanelli et al. [57,65] forecasted prices for FCR, relevant to 3. No works have addressed item 4, penalty forecasting, although such a forecast could be constructed by combining forecasts for 1b and 2.

A limited body of research for machine learning applications addresses PFR directly. Giovanelli et al. [57] assesses the use of a classic neural network and [65] performs a comparative assessment of the performance of non-neural network approaches, namely GBDT (Gradient Boosting Decision Tree), SVR (Support Vector Machine), Linear Regression, and Regression Tree. However, there is no direct comparison between neural network and non-neural network-based approaches in this context. The machine learning solutions presented in this article are provided for exemplary purposes and are similar to the approach presented in [57]. Other more advanced machine learning approaches in the broader field of energy forecasting are potential areas for further research in PFR-related forecasting applications. Liu et al. [66,67] address a significant shortcoming in the said articles, namely a systematic method for determining the significance of each input feature on the forecast quality; the approach could be applicable to the present article for an example related to selecting the key weather features from a large set of available weather features. The machine learning research on PFR is currently focusing on short term forecasting, but further work on long term forecasting with neural networks could benefit from model migration techniques [68].

As virtual power plants manage resources with unpredictable behavior, such as electric vehicle batteries, the forecasts would ideally be accompanied with an uncertainty metric. Kempitiya et al. [69] propose such a metric for PFR price forecasting as well as bidding logic that exploits the forecasts as well as the uncertainty metrics to achieve increased revenues from the PFR markets. Other uncertainty metrics could also be applied from the literature in machine learning solutions for energy forecasting. Gaussian process regression has been applied to determine confidence intervals for the predictions [70–72]. A topic for further research would be PFR market bidding algorithms that are able to utilize the

confidence intervals to maximize market revenue while minimizing the risk of failing to deliver the offered capacity.

### 2.5. Northern European Primary Frequency Reserves Markets: Current Status and Future Trends

The Nordic countries are Finland, Denmark, Iceland, Norway, and Sweden. Due to its geographical isolation, Iceland is not part of the Nordic electrical power markets, though. The remaining countries have evolved a so-called Nordic markets model or power market structure, see e.g., [73], which consists of:

1. Financial market (Nasdaq Commodities), from 10 years down to 1 day ahead of the physical delivery of the power
2. Day-ahead markets (aka Elspot in case of Nord Pool), 1 day ahead of physical delivery
3. Intraday markets (aka Elbas similarly), on the day of physical delivery, and
4. Reserve and balancing power markets, elaborated below.

Finally, imbalance settlement is done on Nordic level post-delivery, to account for differences between plans, forecasts, or trades and actual deliveries. Denmark will fully join this Nordic settlement activity by 2021 [74].

Currently, the time resolution of the electrical power markets in the Nordics is hourly, so the trading, delivery, and settlement will be distinct for every hour of the day (see e.g., [74–77]).

The Nordic reserve and balancing power markets, a.k.a. ancillary services markets, are currently the following throughout the region [78–81]:

1. Fast frequency reserve (FFR), a national day-ahead market in select seasons
2. Frequency containment reserve for normal operation (FCR-N), national yearly, and day-ahead markets
3. Frequency containment reserve for disturbances (FCR-D), mainly national yearly and day-ahead markets (the Transmission System Operator (TSO) may also purchase capacity from neighbouring countries)
4. Automatic frequency restoration reserve (aFRR), a Nordic hourly market in select hours
5. Manual frequency restoration reserve (mFRR), a Nordic hourly market

The time resolution of all the reserve and balancing power markets in the Nordics is also hourly, and it will become quarterly, as with the Nordic power markets [82]. This is related in part to European harmonization of energy markets.

From a grid balancing point of view, the EU's whole Internal Energy Market is split up into different synchronous areas. Synchronous areas are defined as 'areas covered by synchronously interconnected TSOs, such as the Nordic synchronous area' [83]. Synchronous areas are mainly important for the fastest types of reserves, in Europe called Frequency Containment Reserves (FCR), which are dimensioned and operated at this scale [84]. The main task of the FCR is to dampen/stop a sudden drop or rise in system frequency.

The international concept of primary frequency reserves corresponds with the European concept of FCRs. In the Nordic synchronous area, the primary frequency reserves cover the FFR, FCR-N, and FCR-D markets. The main features of these markets are summarized in Table 1.

### 2.6. Summary

According to the research that has been reviewed in this section, a significant number of smart solutions are being developed to equip distributed energy resources with a PFR participations capability. An analysis of the PFR market specifications reveals that the minimum bid size in MW significantly exceeds the capacity of most distributed energy resources, so a solution such as a virtual power plant is needed to aggregate the resources and meet the minimum bid size. Key tasks related to bidding, such as forecasting the available capacity, require data from the virtual power plant. There is a lack of research on virtual power plant information architectures, data model, and application programming interfaces, and this is an obstacle towards applying the emerging body of literature on PFR

provided by distributed energy resources. The contribution of this article is to overcome this obstacle and to encourage further research in the area, possibly resulting in eventual standardization of interfaces and data models.

**Table 1.** Main features of the Nordic primary frequency reserve markets [79,85–92].

|  | FCR-N | FFR | FCR-D |
|---|---|---|---|
| Market area | National | National | National |
| Bid submittal by | 18:30 (EET) | 18:00 (EET) | 18:30 (EET) |
| Minimum capacity | 0.1 MW | 1 MW | 1 MW |
| Maximum capacity | 5 MW per bid | 10 MW (13.5 MW) per bid | 10 MW per bid |
| Control capability | Linear or piecewise linear | Single step activation | Linear, piecewise linear or single step activation |
| To begin from | 0.01 Hz deviation | −0.3 or −0.4 or −0.5 Hz deviation | −0.1 Hz deviation (or see spec re: single step activating assets) |
| Activation speed | Max 180 sec where 0.1 Hz | Max 0.7 or 1.0 or 1.3 s respectively | Max 5 sec 50% of cap., max 30 s 100% where −0.5 Hz |
| Activation duration (sustain) | Continuous (at least 30 min) | Min 5 sec or 30 sec, max as long as deviation > −0.2 Hz | Continuous (at least 30 min) |
| Deactivation | When deviation < 0.01 Hz | Max 20%/sec or not limited, set by min Activation duration | When deviation < 0.1 Hz for 3 min |
| Recovery wait | N/A | Min 10 sec | N/A |
| Recovery power | N/A | Max 25% of capacity, earliest 10 s from end of Deactivation. | N/A |
| Re-activatable | Any time during delivery period | In max 15 min | In max 15 min (or see spec re: energy storages) |
| Direction | Symmetric | Upregulation only | Upregulation only |
| Example technologies | Industrial process, heating, lighting | Electrolysis, EV batteries, UPS | Industrial process, heating, lighting |
| Price formation | Margin price principle | Margin price principle | Margin price principle |
| Energy fee/cost | Energy fee per mFRR price | No energy fee received | Energy fee per mFRR price |

## 3. Architecture and Data Model for a Generic VPP Application Programming Interface

### 3.1. Architecture

An overview of the architecture is presented in Figure 1. The generic VPP API is built on top of the underlying commercial VPP to encapsulate its proprietary details. For security reasons, neither the VPP nor the generic API are exposed to a public network. The communication to frequency reserve markets is opened only for those VPPs that have an agreement with the TSO.

VPP client interfaces between the VPP and the REST API are described in Section 3.3. The client accesses whatever proprietary interface that is provided by the VPP to get the data. It converts the data to comply with the REST API specification and passes it to the REST API, and vice versa for the data flow from the generic API to the VPP. This provides the possibility to use the same generic API with any VPP. Only the VPP client needs to be built and configured for each different VPP.

Generic VPP API consists of a relational database, a REST API, and preprocessing. The relational database is used to persist the data retrieved from the VPP, and its data model is presented in Section 3.2. The REST API provides endpoints to create, retrieve, update, and delete all the necessary data. The REST API also provides the ability to retrieve extensive time series datasets that can be used in the capacity forecaster and to provide data for reports and analytics. Preprocessing is used to create those datasets. It unites multiple timeseries into one table and resamples the data to a specified frequency. The preprocessing can also calculate minimum, maximum, and average values as well as sums, differences, and multiplications. Examples of these calculations are presented in Section 4.1. The REST API is explained in more detail in Section 3.3.

Market forecaster is used to forecast the price of PFR reserves for each market interval of the upcoming bidding period. An applicable solution has been presented in previous research [57], based on supervised machine learning and using open data sources to teach the model and to make the predictions.

Capacity forecaster is used to predict the available capacity that the VPP can use to participate in PFR markets. Capacity in this context is according to the definition in Section 2.4. There is a lack of previous research for this purpose, but a solution is presented in this article, based on supervised machine learning. Relevant data shall be retrieved from the VPP through the VPP client to teach the model and make the predictions. Open data sources can be used to enhance the accuracy of the model; for example, solar radiation data from the local meteorological institute is relevant for predicting the capacity of photovoltaic reserve resources participating on PFR markets.

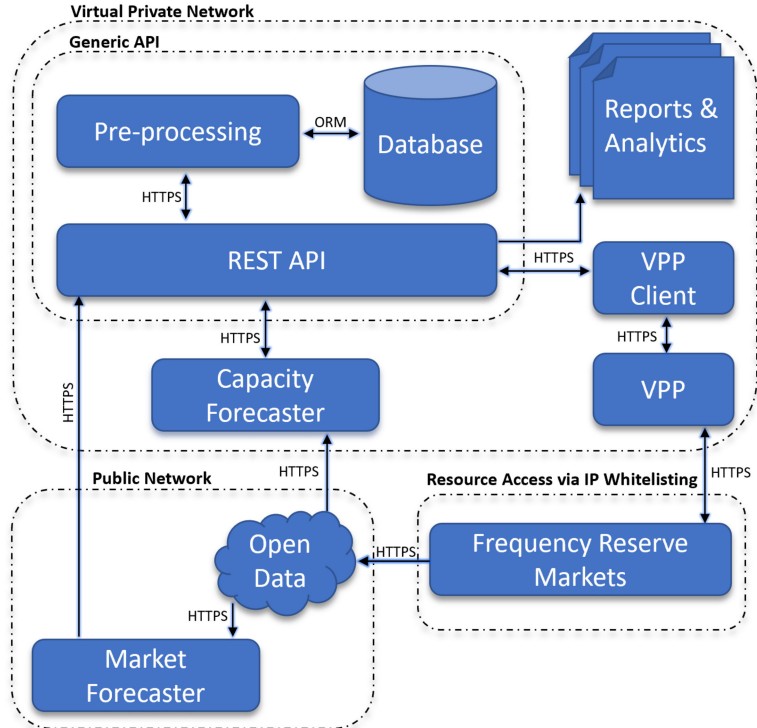

**Figure 1.** Overview of the architecture of the proposed generic VPP API.

### 3.2. Data Model

The data model is described in the Unified Modeling Language (UML) class diagram in Figure 2. Each class has a corresponding table in the database and each attribute has a corresponding column in the table.

Datapoint is used to store timeseries data. Each datapoint has a timestamp field and value field. It also has a foreign key to measurement type and measurement point.

MeasurementType is used to store the metadata of all the different types of timeseries data that are stored. The main types of timeseries are forecasts and measurements. Each MeasurementType has a reference to the identification information the MeasurementType has in the VPP. The field data_after specifies the starting date of the cached data.

MeasurementPoint is a super class for Asset and ControlArea. This allows use of the same DataPoint model to store both asset- and control area-related time series data. Each MeasurementPoint has a reference to the identification information the MeasurementPoint has in the VPP.

Asset is used to store the metadata of all different assets that have the capability to function as reserve resources on the PFR markets. Examples of such assets were reviewed in Section 2.2.

ControlArea is used to store the metadata of all different control areas. A control area pools a set of assets that can be aggregated to a single offer to a frequency reserve market. Each ControlArea has a reference to the identification information the ControlArea has in the VPP. It is also used to maintain an up-to-date list of Markets the ControlArea is used to participate in.

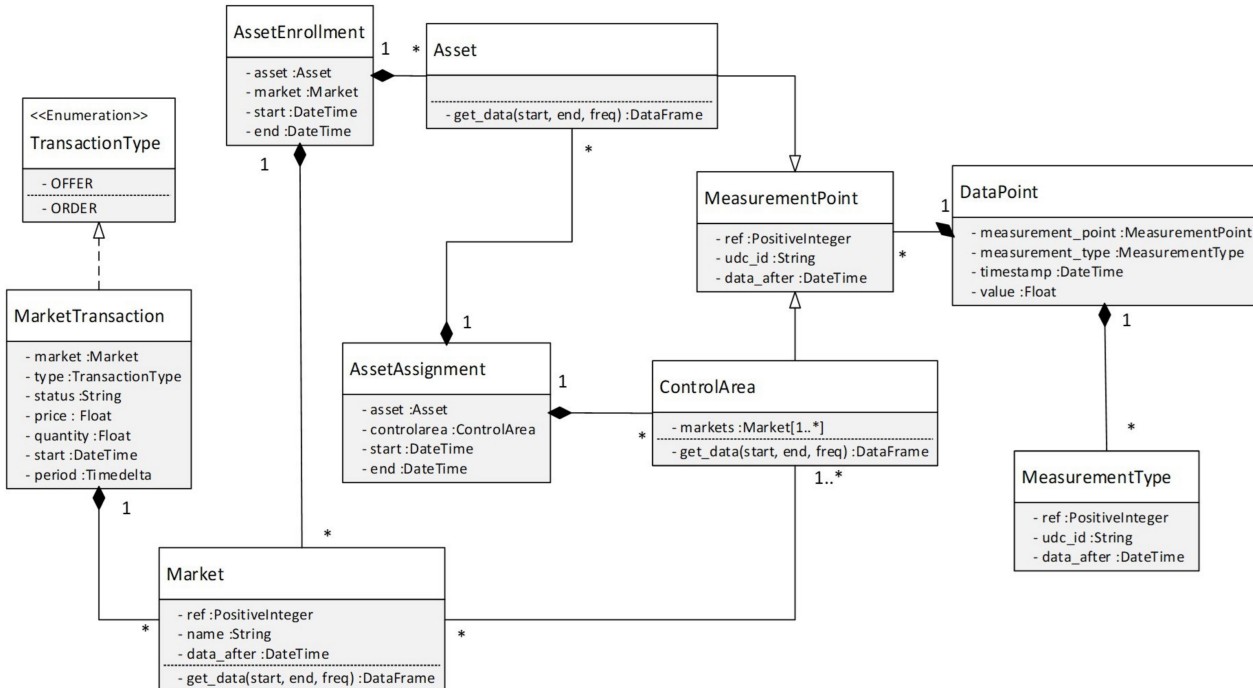

**Figure 2.** UML Class diagram of the data model.

Market is used to store the metadata of different frequency reserve markets. Each Market has a reference to the identification information that the Market has in the VPP.

TransactionType is an enumeration with two possible values: 'offer' or 'order'. Offer is a term that VPP systems use to refer to the bids discussed in Section 2.4. Order is an offer that has been accepted by the market operator.

MarketTransactions is used to store the time series data of offers and orders of different markets. In this context, offers are bids made to a frequency reserve market and orders are bids that the TSO has accepted. Each MarketTransaction has market, type, status, price, quantity, start, and period fields. Market field is a foreign key to a Market. Price field is as defined in Section 2.4. Quantity in this context is the capacity defined in Section 2.4. Start field is the start time of the bid. Period field is the bidding period as defined in Section 2.4.

AssetEnrollment is used to maintain an up-to-date list of assets that are able to participate as reserve resources in a given market at a given time. Each AssetEnrollment has a start and end time and foreign key references to Market and Asset.

AssetAssignment is used to store the time period when a specific asset participated in market activation in a specific ControlArea. Each AssetAssignment has start and end times and foreign key references to ControlArea and Asset. Activation in this context is as defined in Section 2.4. The assignment information is needed for reporting to the TSO.

### 3.3. REST API

REST has become a well-established technology for implementing Information Technology (IT) systems according to the Service-Oriented Architecture paradigm (e.g., [93,94]), and it is used broadly in commercial VPPs, such as the VPP in our case stsudy. For each class in Figure 2, with the exception of the superclass MeasurementPoint, the following REST API endpoints are defined. This is an example for the Asset class:

- GET/assets/
- GET/assets/{ref}/
- GET/assets/{ref}/data
- POST/assets/
- PATCH/assets/{ref}/
- PUT/assets/{ref}/

- DELETE/assets/{ref}/

There are three GET endpoints. The first one lists all the assets configured into the VPP. The following is an example of the JSON (JavaScript Object Notation) output in the case of a VPP with two assets:

```
[ {
"udc_id": "CHP1",
"sub_type": "Generation",
"active": true,
"id": 1,
"type": "VPP",
"ref": 51,
"status": "Active",
"data_after": "2020-06-03 11:45:00"
}, {
"udc_id": "Building1",
"sub_type": "Load",
"active": true,
"id": 2,
"type": "VPP",
"ref": 52,
"status": "Active",
"data_after": "2020-06-03 11:45:00"
} ]
```

The second GET returns the unique asset with the unique id 'ref' (ref is a field of Asset's superclass MeasurementPoint in Figure 2.) The third GET is only available for those classes in Figure 2 that have timeseries data, namely Asset, ControlArea, and Market. It returns the timeseries for all of the MeasurementTypes associated with the said class.

There are three endpoints that are used to modify the assets: POST, PATCH, and PUT. The POST endpoint is used to add a new asset. PATCH and PUT endpoints are used to update the assets. The PUT endpoint requires that data for all fields of an asset are submitted; with the PATCH endpoint, it is possible to submit a partial modification to one or more fields.

The DELETE endpoint is used to delete an asset.

## 4. Use Cases

The applicability of the proposed virtual power plant application programming interface is demonstrated with two use cases.

The first use case is the operational reliability of an asset. Its objective is to determine an indicator that captures how well an individual asset has delivered its forecasted PFR capacity. This is of interest to several actors. Firstly, a maintenance manager is able to target maintenance activities to assets that score poorly on this indicator. Secondly, the trader on PFR markets is able to use this information to assess the likelihood of the assets failing to deliver the forecasted capacity, and to take this into account when deciding the capacity to bid on the PFR market. Thirdly, the virtual power plant operator and the asset owners need to have an agreement about how the revenues from the PFR market are shared. The operational reliability of an asset is one possible metric that can be used to determine the shares. The presentation in Section 4.1 is applicable to any asset managed by the virtual power plant. The case study in Section 6.1 is specific to a battery storage asset.

The second use case is a forecast of the PFR capacity of an asset. Its objective is to deliver forecasts of the capacity for all intervals of the upcoming bidding period. The PFR market in several countries, including the case study country Finland, is day-ahead and hourly, so one capacity forecast is needed for each hour of the next day. The main actor for this use case is the trader, since the trader must specify the capacity in the bid for each hour, as specified in Sections 2.4 and 2.5. The trader's objective is to maximize

the capacity, and thus the revenue, while avoiding penalties related to situations in which the asset is not able to fully deliver the capacity specified in the bid. The presentation in Sections 4.2 and 5 is generally applicable to virtual power plant assets and needs to be adapted to a specific asset type through the selection of relevant input features to the machine learning forecaster. Section 6.2 presents the relevant inputs to the photovoltaic asset capacity forecaster.

### 4.1. Use Case 1: Operational Reliability of Virtual Power Plant Assets

The operational reliability of an asset is calculated in the preprocessing module of Figure 1. The definition of capacity in Section 2.4 is elaborated as follows:

- CapacityNeg: If the asset is a load, this is the adjustable additional power consumption that can be activated in over frequency events. If the asset is a generator such as a PV panel, this is the adjustable curtailable power production.
- CapacityPos: If the asset is a load, this is the adjustable curtailable power consumption that can be activated in underfrequency events. If the asset is a generator, this is the additional power production that can be activated.
- ForecastCapacityPos: This is the forecasted CapacityPos.
- ForecastCapacityNeg: This is the forecasted CapacityNeg.

A special case of an asset is an energy storage. A storage may participate on PFR markets by curtailing its charging power, in which case it behaves in the same way as a load, and the curtailable power is specified by CapacityPos; in case the storage is able to increase its charging power in an overfrequency event, the possible power increase is specified by CapacityNeg. In case the storage is able to provide power to the grid through a grid inverter, it acts as a generator, and the provided power is specified by CapacityPos.

The third GET in the REST API in Section 3.3 can be used to obtain the timeseries for all of the MeasurementTypes associated with the Asset class, including the types in the above list.

In commercial VPPs, the operational reliability of an asset is a metric of how well it actually delivers its forecasted capacity. It is defined as follows:

Min(CapacityPos, CapacityNeg)/Min(ForecastCapacityPos, ForecastCapacityNeg)

### 4.2. Use Case 2: Reserve Capacity Forecasting

An advanced feature of a virtual power plant is to forecast the ForecastCapacityPos and ForecastCapacityNeg timeseries (defined in Section 4.1). Such forecasts are useful for the VPP when it bids on the frequency reserves markets. Since the CapacityPos and CapacityNeg timeseries (defined in Section 4.1) are available through the REST API, a supervised learning type of machine learning forecast is possible. This approach will be applicable to all kinds of reserve resources managed by the VPP, which have been prequalified for the frequency reserves markets. Such reserves include PV, batteries, electric vehicles, and building automation loads.

## 5. Implementation

The box with the label Capacity forecaster in Figure 1 is implemented as follows. Supervised learning is applied to the time series forecasting problem defined in Section 4.2. The approach is similar to [57], but without the restrictive assumption that the AI model is trained once per day and then used to make predictions for each hour of the next day. The goal is to provide predictions for a time series at intervals called epochs. In the results section of this paper, the epoch is 1h, which is the bidding interval of frequency reserves markets in Finland.

In supervised learning, the model is trained with a labelled set of training samples. In time series forecasting, there is one sample for each epoch. A matrix $X_{train}$ is constructed with one row for each feature and one column for each epoch. The labels are the correct values for the time series to be predicted for the said epochs and they are stored in a one-column matrix $Y_{train}$. Since the goal is to obtain the values for the ForecastCapacityPos

and ForecastCapacityNeg timeseries (defined in Section 4.1), the labels are CapacityPos and CapacityNeg, respectively (defined in Section 4.1).

The most recent epoch in the training set cannot be any more recent than the most recent epoch for which the label would be available, if the solution would be deployed in real time. When $X_{train}$ and $Y_{train}$ are available, it is possible to train an AI model. The model architecture and hyper-parameters are based on time series forecasting research in this domain [57]. A three-layer classic neural network is used. Both hidden layers are followed by a 'dropout' layer for the purpose of regularization, i.e., preventing the model from overfitting to the training samples. Table 2 provides additional information for AI practitioners who may wish to reproduce the results. The Adam optimizer is used to ensure the convergence of neural network training.

**Table 2.** Neural network architecture and hyper-parameters.

| Layer | Type | Number of Nodes | Dropout | Activation Function |
|---|---|---|---|---|
| 1st hidden layer | Dense | Same as input layer | 0.4 | tanh |
| 2nd hidden layer | Dense | 24 | 0.4 | tanh |
| Output layer | Dense | 1 | No dropout | relu |

The purpose of AI is to obtain the values for a set of future epochs for the time series to be predicted. This is a column matrix $Y_{pred}$. Thus, a matrix $X_{pred}$ is constructed, which is otherwise similar to $X_{train}$ but has the values for the said future epochs. Thus, all features must be selected in such a way that data are available for those epochs. If this is not the case, the following feature engineering technique can be used: a time shift is applied, such as using the data from the day before. The same time shift applies to $X_{pred}$ and $X_{train}$.

Figure 3 shows the data flows. $X_{train}$ and $Y_{train}$ are used to fit the model. $X_{pred}$ and $X_{train}$ are concatenated before normalization, so that the training data and $X_{pred}$ have the same statistical properties. In our implementation, X had features in rows and $Y_{pred}$ is a column vector, so X is transposed before model fitting. $X_{pred}$ is given as an input to the fitted model, which outputs the forecasts $Y_{pred}$. Since the research is conducted with historical data, the actual values $Y_{real}$ were also available and have been plotted in the results section alongside the forecasts.

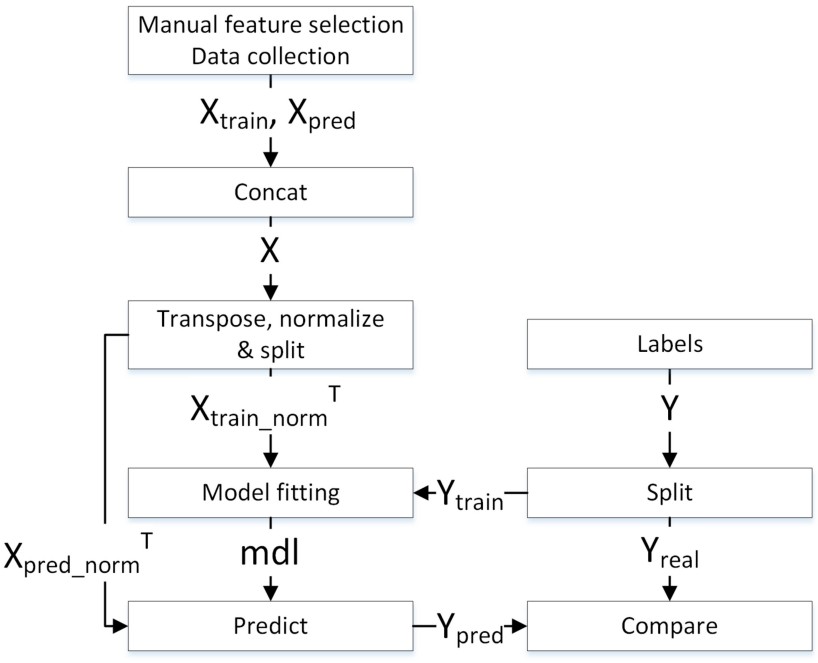

**Figure 3.** Data flows for training the machine learning model and using it to make forecasts.

The implementation of the *Database* in Figure 1 is as follows. Django is a high-level Web application development framework broadly used in industry and academia (e.g., [95,96]). It has been used to implement the data model in Figure 2. This means that the data model consists of multiple *Django models*. A *Django model* is the single, definitive source of information about the data. It contains the essential fields and behaviors of the stored data. Each model maps to a single database table and each model is a Python class that subclasses django.db.models.Model.

In use case 1, the ForecastCapacityPos, ForecastCapacityNeg, CapacityPos, and CapacityNeg are obtained from the virtual power plant through our API. In use case 2, the matrix Y data is likewise obtained from the virtual power plant. The matrix X uses weather data from the relevant region as input features to the neural network; the data is obtained from the Finnish Meteorological Institute open API https://en.ilmatieteenlaitos.fi/open-data-sets-available (accessed on 22 February 2021). The trading interval for PFR in the case country Finland is hourly, so the datasets have one datapoint per hour. Further research may utilize additional data from the control systems of the distributed energy resources. In the case of sensor data from control systems, there will be a large number of data points per hour, so these need to be preprocessed to obtain one data point per hour. One direction of further research is preprocessing solutions that intelligently filter the sensor noise.

## 6. Case Study

### 6.1. Use Case 1: Operational Relisability of Virtual Power Plant Assets

Figure 4 shows the time series CapacityPos, CapacityNeg, ForecastCapacityNeg, and ForecastCapacityPos, defined in Section 4.1, and the Operational reliability which is calculated based on these time series by the equation in Section 4.1. These series are from a battery storage unit in a shopping mall. The unit is participating on the Finnish FCR-N market and the data is obtained through the VPP API of the VPP that manages the energy resources of the shopping mall. In Figure 4, ForecastCapacityNeg and ForecastCapacityPos both have a constant value of 1.7 MW. Thus, based on the equation in Section 4.1, it is expected that the Operational reliability will look like min (CapacityPos, CapacityNeg) and be scaled by dividing by 1.7 MW. This behavior is observed in Figure 4.

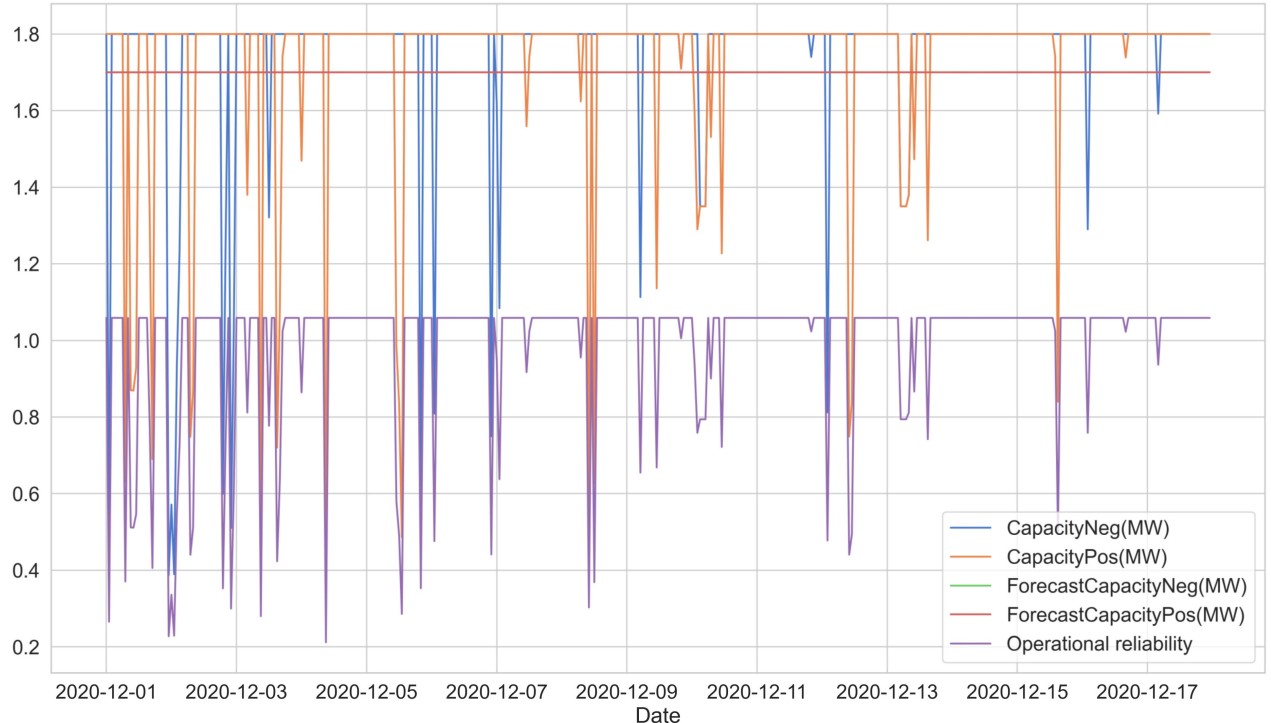

**Figure 4.** Operational reliability calculation for a battery storage asset.

### 6.2. Use Case 2: Reserve Capacity Forecasting

Reserve capacity forecasting of a photovoltaic PFR resource is performed on the data received from the VPP application programming interface. This operational VPP is installed at a Finnish shopping center, aggregating photovoltaic and other PFR resources. This use case mainly targets to forecast a day-ahead photovoltaic capacity for each hour of the next day, i.e., forecasting *ForecastCapacityPos* and *ForecastCapacityNeg* time-series (defined in Section 4.1) from the labels *CapacityPos* and *CapacityNeg*, respectively (defined in Section 4.1). For this purpose, historical data of about 16 months is collected from the VPP application programming interface.

The response from the application programming interface includes the PV capacity (in 10 kW) data for every 15-min interval. The forecasting is targeted for every hour, but there are more data points than required. The preprocessing is done by taking the average of the data points per hour. Figure 5 shows the preprocessed PV data collected from the application programming interface. In addition to VPP data, openly available solar radiation data is collected from the local meteorological institute. Utilizing the open API of the Finnish Meteorological Institute, data is collected for every hour for 16 months in the same time frame as the PV capacity data. This open API provides the solar radiation features listed in Table 3.

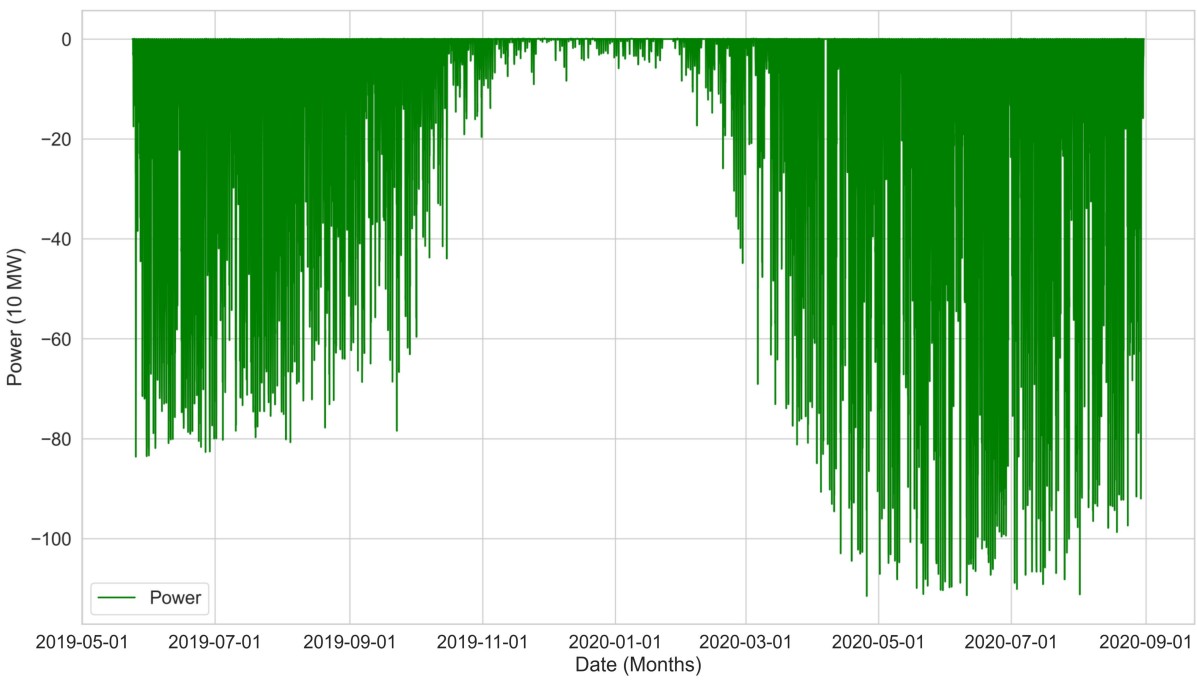

**Figure 5.** Aggregated FCR-N capacity of the photovoltaic panels at the shopping center.

**Table 3.** Solar Radiation features provided by open API.

| Parameter | Unit |
|---|---|
| Diffuse radiation | W/m$^2$ |
| Direct solar radiation | W/m$^2$ |
| Global radiation | W/m$^2$ |
| Long wave solar radiation | W/m$^2$ |
| Long wave outgoing solar radiation | W/m$^2$ |
| Radiation balance | W/m$^2$ |
| Reflected radiation | W/m$^2$ |
| Sunshine duration | s |
| Ultraviolet irradiance | index |

Among these features, except Direct solar radiation (W/m$^2$), Radiation balance (W/m$^2$), and Ultraviolet irradiance (index), all other features were chosen for the model training. Figure 6 portrays these six features plotted alongside with the PV power output.

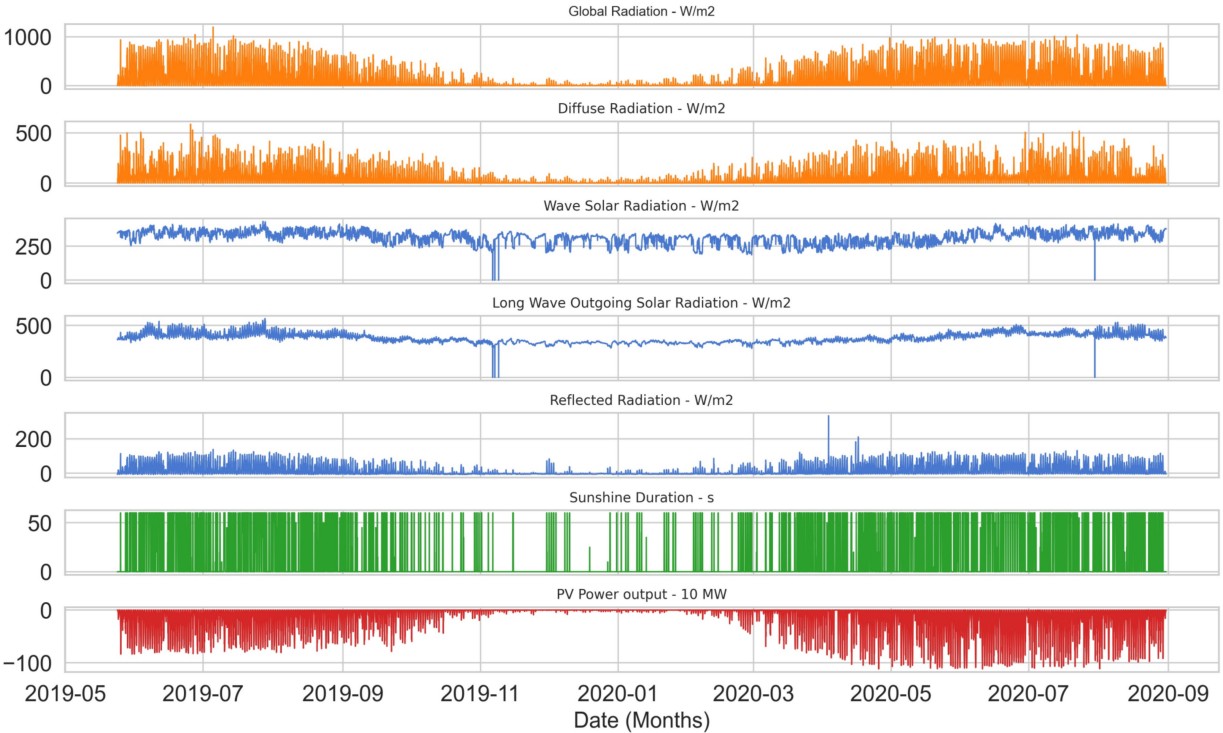

**Figure 6.** FMI radiation observation data.

Figure 6 shows an apparent influence of solar radiation features on the PV power output. For instance, like other features, the feature 'Global radiation' peaks from May 2019 to August 2019. In the same time frame, the power output from the PV panel is also high. Conversely, from October 2019 to March 2020, the feature value and power output both are low. Furthermore, a steady increase or decrease in the power values are directly proportional to the steady increase or decrease in the feature values. From this analysis, it is evident that these six features are relevant with respect to the power generation in the PV panels, and they can be used in forecasting.

Additionally, after analyzing the power output patterns from Figures 7 and 8, the data seems to have a vital seasonality component. For example, the PV power output is high between 10 and 16 h. Similarly, the month around July seems to contribute more to the power generation and can provide more data to the network by explicitly referencing days and months. Thus, these calendar features have been included.

The implementation method, model architecture, and hyper-parameters of PV capacity forecasting are mentioned in Section 5. The VPP application programming interface and meteorological institute's collected data are aggregated to a single data source and the features are manually selected. The matrix $X_{train}$ is constructed with one row from each of the selected features and one column for each epoch, i.e., hour, as specified in Section 5. Once the forecasts were run by the system in Figure 3, the results were plotted, as shown in Figure 9. In this use case, as the research is carried with historical data, $Y_{real}$ was also available along with $Y_{pred}$ from the forecast results. These three parameters: Historical data, $Y_{real,}$ and $Y_{pred}$ were plotted alongside one another in Figure 9.

Figure 10 shows True PV capacity (i.e., $Y_{real}$) vs. Predicted PV capacity (i.e., $Y_{pred}$) to get more clarity on the forecast results. By comparing the true and predicted values, it can be observed that the model is performing very well in forecasting the PV capacity values. However, few outlier power patterns were observed in some hours. For instance, from

Figure 11, the day's starting hours have been accurately predicted by the model, but one outlier pattern is observed for a day. This pattern can be clearly seen in Figure 12.

Overall, this use case's objective was to predict the day-ahead PV capacity from the VPP application programming interface data. This research described the problem analysis and implementation scheme used to forecast the capacity forecasting.

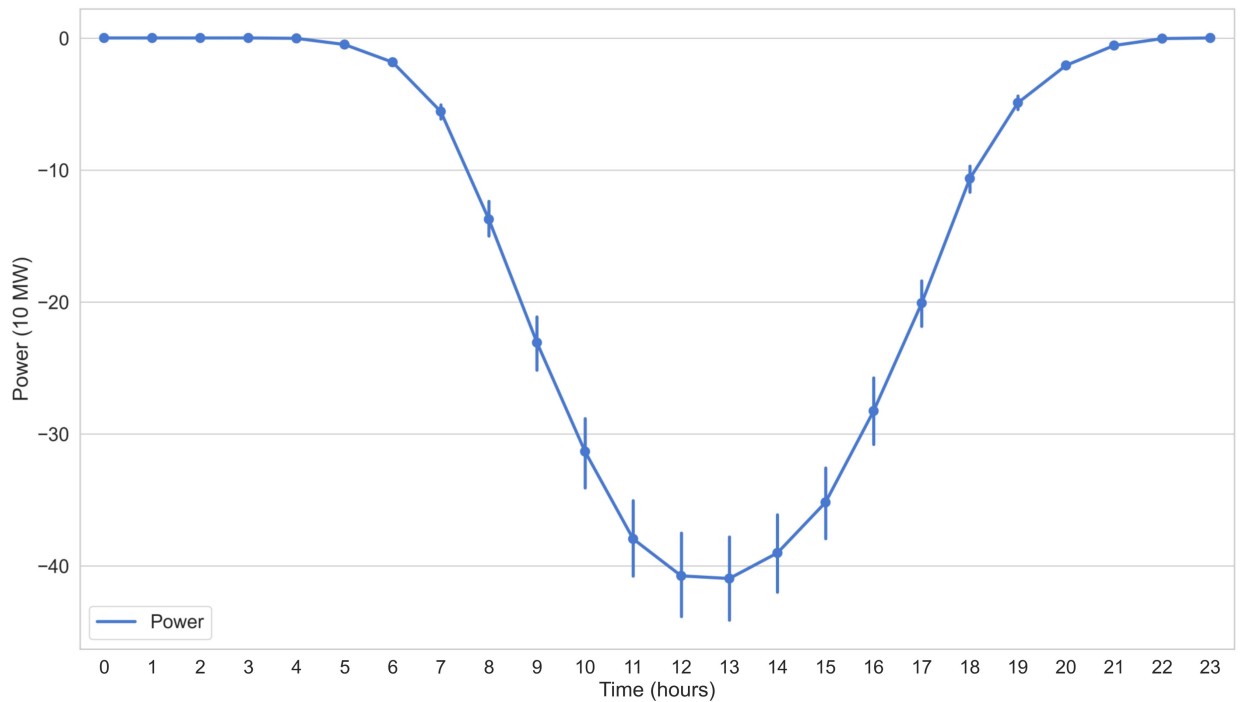

**Figure 7.** Hourly power output from Photovoltaic Panel.

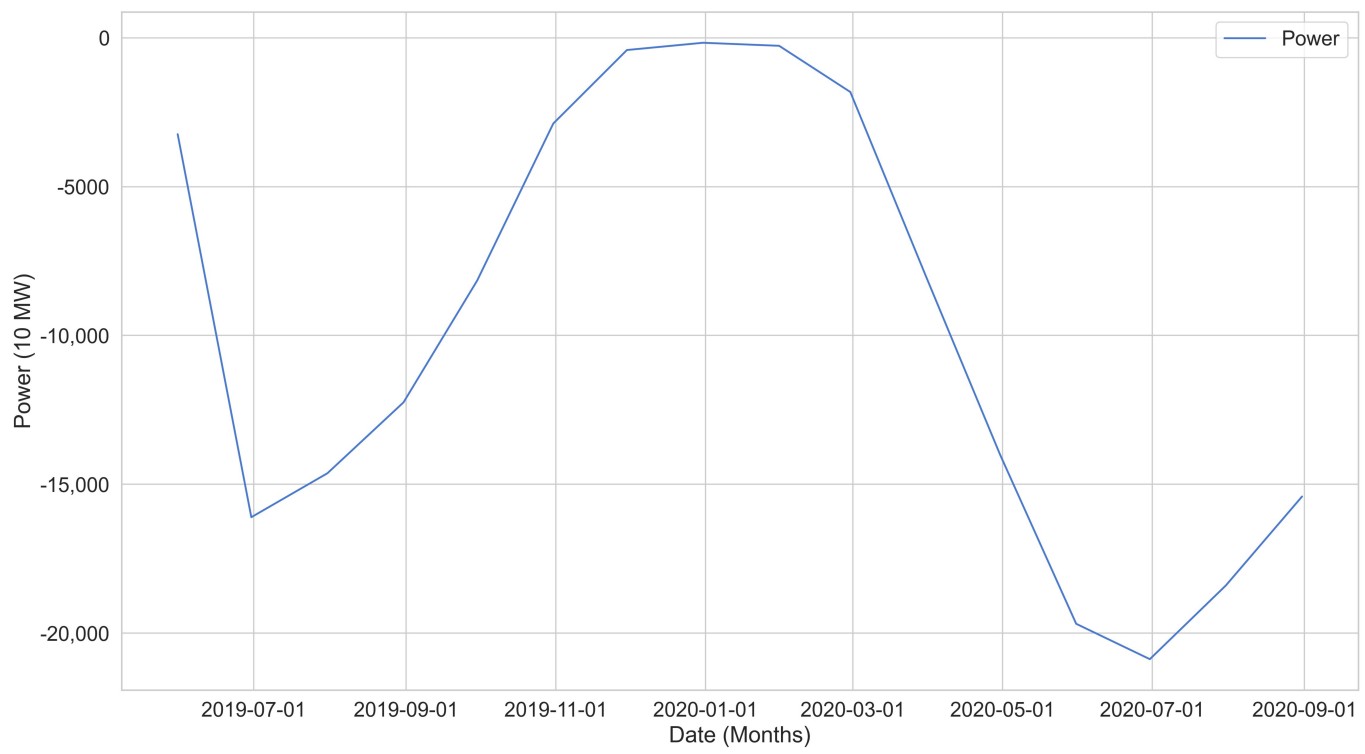

**Figure 8.** Monthly power output from Photovoltaic Panel.

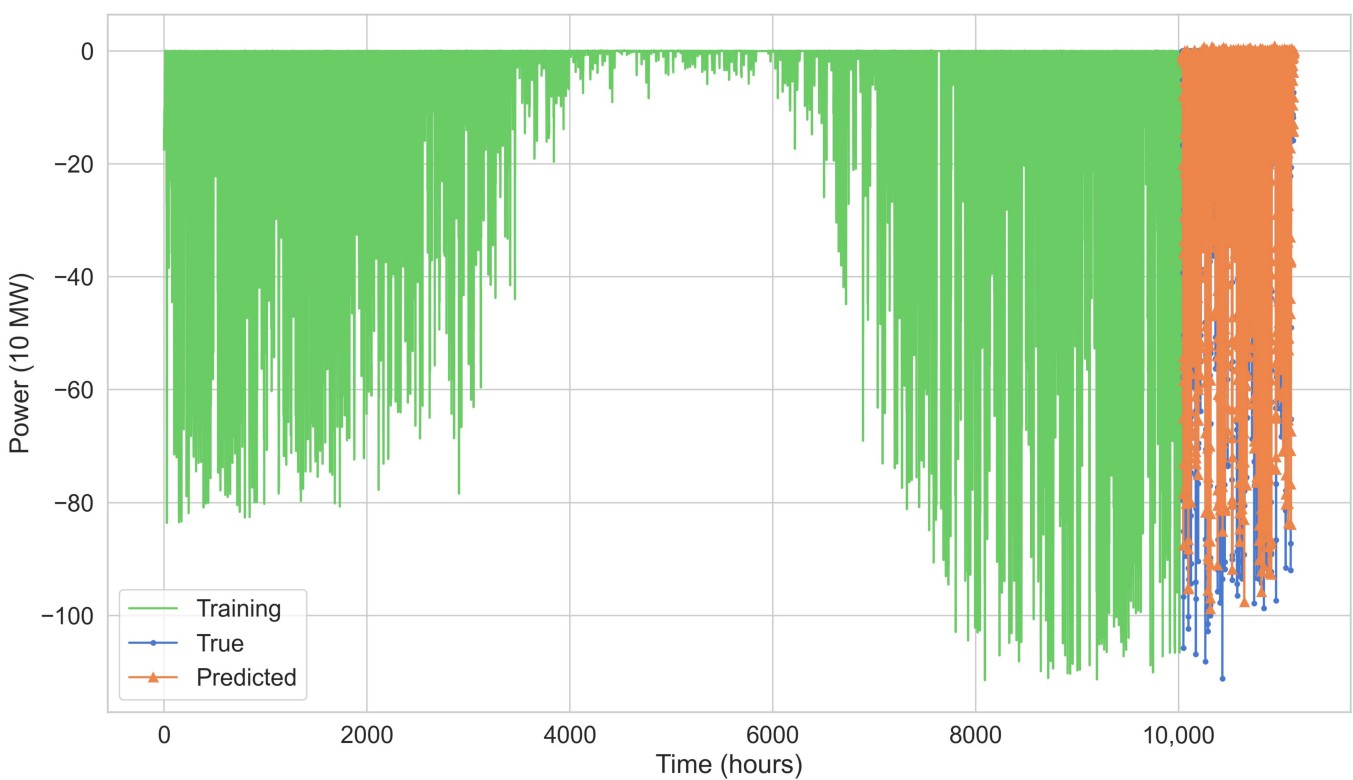

**Figure 9.** True vs. Predicted PV capacity output with training data.

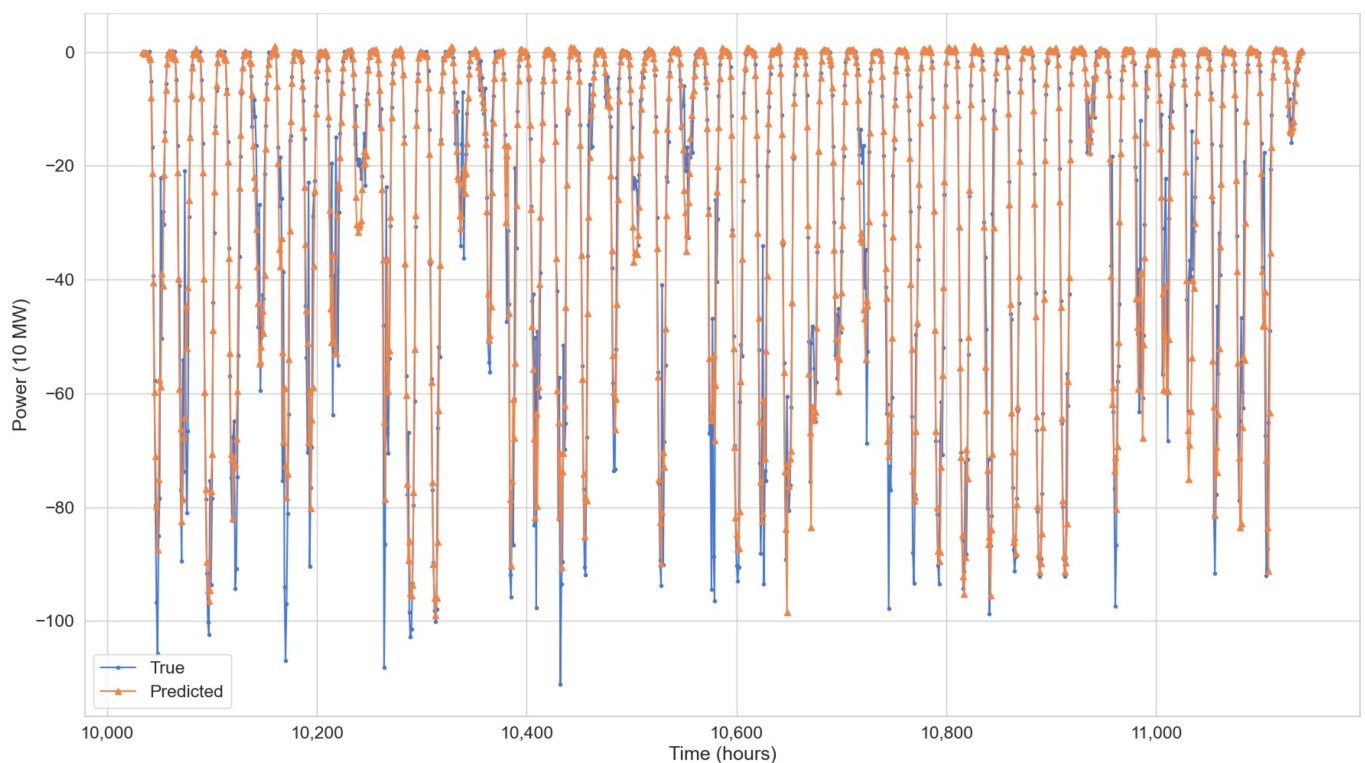

**Figure 10.** True PV capacity vs. Predicted PV capacity.

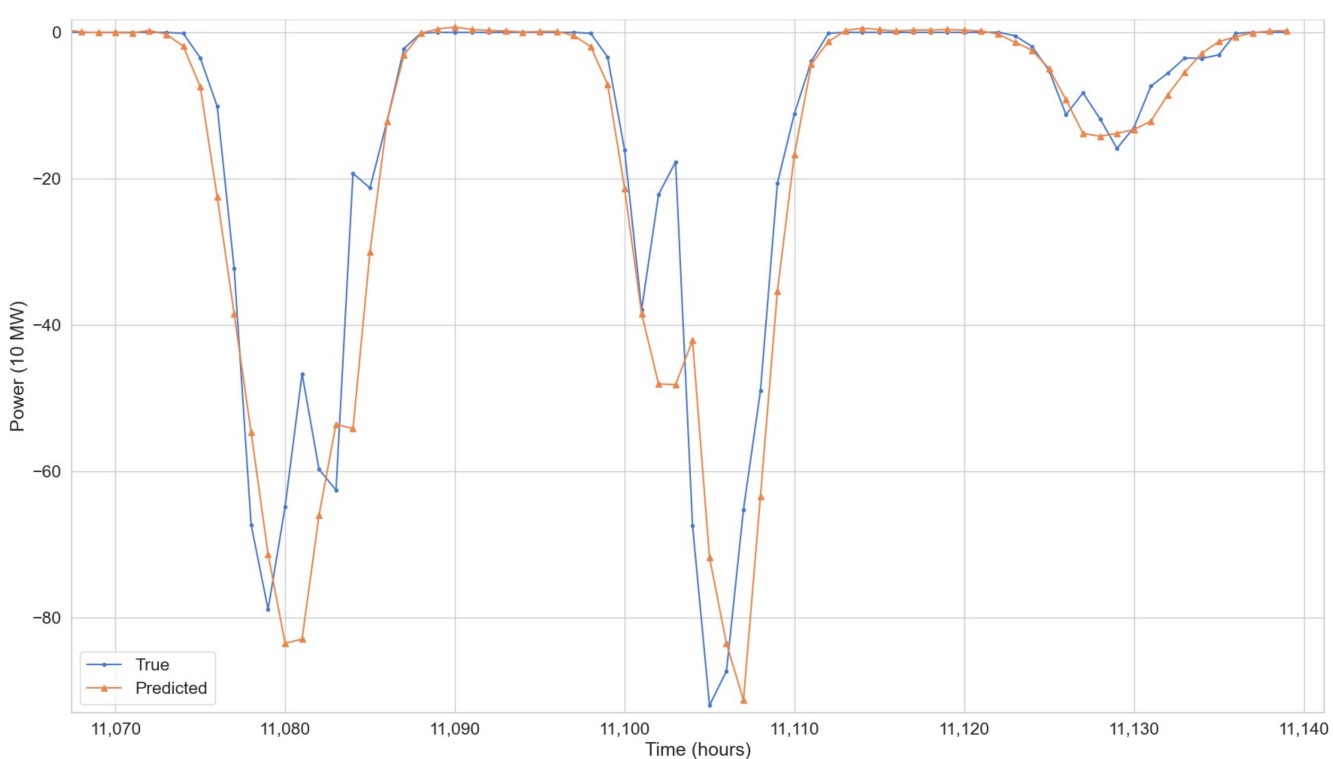

**Figure 11.** True vs. Predicted PV capacity output for 3 days.

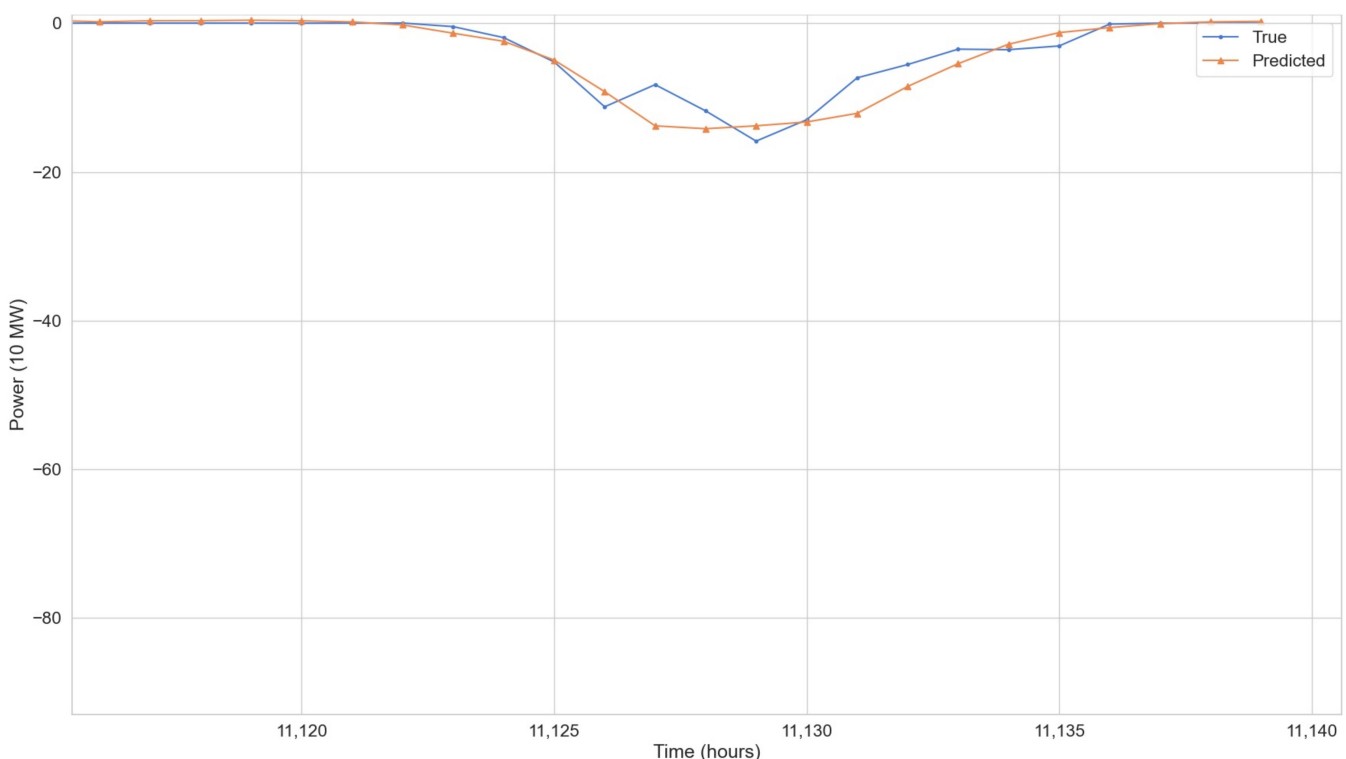

**Figure 12.** True vs. Predicted PV capacity output for 1 day.

## 7. Discussion

The case studies demonstrate application scenarios of the virtual power plant architecture, data model and application programming interface. Both case studies exploit time series data from the virtual power plant as source information in order to perform analytics

and to generate a new time series of interest to the end user. From Figure 2 and the REST API calls in Section 3.3, it is clear that a commercial virtual power plant does not provide user-friendly interfaces, which energy domain experts can be expected to use to request the source information. Figure 2 is understandable for a computer science expert and can serve as the basis for multidisciplinary collaboration between such professionals as well as energy sector professionals building advanced features to virtual power plants. Both of the use cases presented in this article are examples of such advanced features. Once the source information has been obtained and the advanced feature has been implemented, the result is a new dataset, such as the operational reliability of an asset or the predicted PV PFR capacity. There remains the question of how such a dataset can be used. This question is especially relevant in the context of virtual power plants that visualize all relevant data for users and may even perform automatic trading on PFR markets. Thus, it is necessary to input the result datasets to the VPP in a format in which it is able to receive it. From Figure 2, it is seen that both of the mentioned result datasets are a series of datapoints associated to a specific asset, so one of the endpoints in the REST API, namely POST, PATCH, or PUT can be used to write the results back to the virtual power plant's database, after which they can be exploited by the virtual power plant.

It is notable that the work presented in this article is based on one commercial virtual power plant. According to the experience of the authors, the material in Section 3 would be applicable to other vendors' virtual power plants. Unfortunately, there is a lack of academic as well as professional publications on the subjects. This is an obstacle for researchers who would like to perform case studies with real power plants. It is also an obstacle for industry seeking to exploit research in this area by interfacing innovative algorithms as advanced features to their power plant. The authors welcome follow-up research by other groups working with different virtual power plant vendors, aiming at an internationally shared vision of virtual power plant architecture, data models, and application programming interfaces.

## 8. Conclusions

This paper has focused on the following gap in VPP research: Although much research is being published, its applicability in the context of real-world VPPs is often unclear. This paper has proposed a generic architecture, data model, and API for accessing real VPPs. Its applicability has been demonstrated in the context of an operational commercial VPP in Finland for two use cases: (1) computing the operational reliability of VPP managed assets, and (2) forecasting the capacity that PV assets can offer on Finnish frequency reserve markets. The first use case is straightforward and serves the purpose of demonstrating the industrial applicability and technology readiness level. The second use case has implications on advanced features of a VPP, namely its ability to bid intelligently to maximize its revenue from trading its assets on frequency reserves markets. The bidding is day-ahead in Nordic markets, and the capacity forecasting solutions work on the same day-ahead timeframe. A key component of the bid is the capacity of adjustable power that VPP is able to provide the next day to react to frequency deviations. The revenue is directly proportional to this capacity. Thus, the VPP should maximize the capacity in its bids while avoiding the penalties due to failing to deliver the promised capacity. The PV capacity forecasts presented in this paper are directly applicable for the bidding module of the VPP.

Several directions of future work are possible. Firstly, the forecasting approach could be applied to other types of assets in addition to PV; in this case, a key task would be to select the relevant features that the machine learning model needs to predict the capacity of that type of asset. Secondly, bidding strategies could be developed on top of the forecasts. Thirdly, further case studies on different commercial VPPs operating in different countries could further validate and develop the generality of the proposed VPP data model and API.

**Author Contributions:** Conceptualization, R.S., M.Y.-O. and S.S.; methodology, R.S., M.Y.-O. and S.S.; software, R.S. and M.Y.-O.; validation, R.S., J.V., S.S., V.V.; formal analysis, R.S.; investigation, R.S.; resources, R.S., M.Y.-O., J.V. and S.S.; data curation, R.S.; writing—original draft preparation, R.S., M.Y.-O., T.H. and S.S.; visualization, R.S.; supervision, S.S., J.V. and V.V.; project administration, T.H. and S.S.; funding acquisition, S.S., T.H. and J.V. All authors have read and agreed to the published version of the manuscript.

**Funding:** This research was supported by Business Finland grant 7439/31/2018.

**Institutional Review Board Statement:** Not Applicable.

**Informed Consent Statement:** Not Applicable.

**Data Availability Statement:** Data is contained within the article.

**Conflicts of Interest:** The authors declare no conflict of interest.

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
