# Peer review of "A Virtual Power Plant Solution for Aggregating Photovoltaic Systems and Other Distributed Energy Resources for Northern European Primary Frequency Reserves"

_energies, doi:10.3390/en14051242_

Round 1

Reviewer 1 Report

Dear Authors,

Thank You for possibility of reading this article.

General statements:

-> Article aims to propose a generic data model and application programming interface for a VPP with the above-mentioned capabilities, as well as, use the application programming interface to cope with the unpredictability of the frequency reserve capacity that the photovoltaic systems and other DERs are able to provide to the frequency reserves markets in the upcoming bidding period. So the aim of the research is actual and desirable.

-> Article suites to Energies journal scope.

-> abstract is well written and is adequate to article content.

-> the literature review is based on 69 positions.

-> the methodology part is clearly introduced.

-> article is organized into 7 sections. The sections are correctly named and ordered

-> quality of figures and tables is sufficient.

However I indicated elements that require revision:

#1 Literature review

After a well-prepared literature review in section 2, there is a need of summarizing them. Then after a short summarize it’s necessary to highlight the contribution of this paper, to assure what research lack is covered by this article.

#2 Discussion

The article presents the case studies, that increase the value of the article. However, the results are not discussed. Thus please add a separate section with a discussion of the results.

#3 Keywords

Please revise keywords. Please revise and assure, they are adequate with article content. For e.g. smart grids are not introduced in any place of the article.

#4 Values and units

There must be a space between value and unit. Please revise in the whole article.

#5 Technical issue

-> reference of table 1 must be changed as it’s guided in the temple. This format is not acceptable.

-> the formatting of the references is totally wrong. It looks like the authors even do not try to follow the temple.

#6 Typos

Typos line 449 “virtual powerplant” should be “virtual power plant”

Reviewer 2 Report

Machine learning-based power plant solution is crucial for aggregating photovoltaic systems and distributed energy resources. It is interesting. Some comments are given below:

1. The introduction should be improved with more state-of-the-art machine learning applications for energy based forecasting. The real contribution of your work in comparison with the existing ones should be highlighted.

2. What are the real benefits and necessity of using neural network in your study? Please clarify them carefully.

3. When describe powerful machine learning for energy based application, please consider these highly related works: Mass load prediction for lithium-ion battery electrode clean production: A machine learning approach; Model migration neural network for predicting battery aging trajectories; Feature Analyses and Modelling of Lithium-ion Batteries Manufacturing based on Random Forest Classification.

4. How to guarantee the convergence of your neural network training and avoid your results being trapped into the local optimum conditions? Please clarify them.

5. Have you considered the uncertainty quantification when design your forecasting solution as uncertainty would highly affect the frequency reserves results? Please carefully clarify this with give evidence.

6. When describe the powerful solutions to achieve uncertainty quantification, please considering these highly related works: Gaussian process regression with automatic relevance determination kernel for calendar aging prediction of lithium-ion batteries; Modified Gaussian process regression models for cyclic capacity prediction of lithium-ion batteries. An evaluation study of different modelling techniques for calendar ageing prediction of lithium-ion batteries.

7. The measurement and shift noises from sensors would highly affect your forecasting results especially for neural network-based method. Have you considered the effects of these noises?

8. The method should be compared with other state-of-the-art solutions to further prove the effectiveness of your applications.

Reviewer 3 Report

This paper proposes a generic architecture, data model and API for accessing real Virtual Power Plant. In general, the paper is interesting, even though several clarifications should be added.

All the following indicated aspects should be clarified and better explained in the manuscript.

Introduction / Literature review

  1. The authors should better highlight the innovative aspects of their work in the manuscript.

System model

  1. Even thought the VPP application programming interface is clearly described, the authors should highlight who are the users and stakeholders of the proposed interface. VPPs affects different level of management platforms, that are closely interconnected and development ground for aggregating photovoltaic systems and other distributed energy resources. However, in the manuscript no indication are reported about how (and/or if) the proposed model is able to interact with:
    1. higher level management platforms such as market area decision support systems as well as those described in the following documents (that could be cited by the authors in the text): https://doi.org/10.1016/j.rser.2015.10.133; https://doi.org/10.1109/TASE.2016.2593101.
    2. Peer-to-peer management platforms such as district energy management decision support systems as well as those described in the following documents (that could be cited in the text): https://doi.org/10.1109/TSTE.2015.2441107, https://doi.org/10.1016/j.apenergy.2017.05.066.
    3. Lower level management platforms such as building/home subsystems and devices SCADA/schedulers as well as those described in the following documents (that could be cited in the text): https:// https://doi.org/10.1109/TASE.2020.2986269; https://doi.org/10.1016/j.apenergy.2016.02.058.

Does the proposed VPP application programming interface take care of all these functionalities?

Use cases:

  1. The authors should elaborate more on the use cases. What are the objectives? An outline about the involved actors, their aims, etc. could also help reader following the whole description.
  2. Use case 1: what about if the asset is an energy storage? In CapacityNeg and CapacityPos, only loads and generators are considered. The authors should update the provided definitions.
  3. Use case 2: is the proposed forecast algorithm novel? is this a contribution of the paper? This contribution seems far from the primary one, which seems to be the VPP software architecture.

Case study

  1. The presentation of numerical experiments could be improved: where do the presented data come from?

Minor

  1. The authors should check that all the used acronyms are explained the first time they are used.
  2. Typos: Line 403: reference to Fig. 1 instead of Fig. 3.

Round 2

Reviewer 1 Report

Dear Authors,

Thank You for the revision. All my propositions were included. Thus I recommend publishing this paper.

Regards 

Reviewer 2 Report

the authors well answered my questions. I think it can be acceptable

Reviewer 3 Report

The Authors’ reply seems to address the raised questions. The updated version of manuscript is much better than the previous one. In the opinion of this Reviewer the manuscript deserves to be published.